# Blind Source Separation in Polyphonic Music Recordings Using Deep Neural Networks Trained via Policy Gradients

**Sören Schulze** [1,*] **, Johannes Leuschner** [1] **and Emily J. King** [2]

1 Center for Industrial Mathematics, University of Bremen, Bibliothekstr. 5, 28359 Bremen, Germany; jleuschn@uni-bremen.de
2 Mathematics Department, Colorado State University, 1874 Campus Delivery, 111 Weber Bldg, Fort Collins, CO 80523, USA; emily.king@colostate.edu
* Correspondence: sschulze@uni-bremen.de

**Abstract:** We propose a method for the blind separation of sounds of musical instruments in audio signals. We describe the individual tones via a parametric model, training a dictionary to capture the relative amplitudes of the harmonics. The model parameters are predicted via a U-Net, which is a type of deep neural network. The network is trained without ground truth information, based on the difference between the model prediction and the individual time frames of the short-time Fourier transform. Since some of the model parameters do not yield a useful backpropagation gradient, we model them stochastically and employ the policy gradient instead. To provide phase information and account for inaccuracies in the dictionary-based representation, we also let the network output a direct prediction, which we then use to resynthesize the audio signals for the individual instruments. Due to the flexibility of the neural network, inharmonicity can be incorporated seamlessly and no preprocessing of the input spectra is required. Our algorithm yields high-quality separation results with particularly low interference on a variety of different audio samples, both acoustic and synthetic, provided that the sample contains enough data for the training and that the spectral characteristics of the musical instruments are sufficiently stable to be approximated by the dictionary.

**Keywords:** blind source separation; policy gradient; neural network; dictionary learning; parametric model; unsupervised learning

## 1. Introduction

We address the problem of unmixing the contributions of multiple different musical instruments from a single-channel audio recording. We assume that each instrument only plays a single musical tone at a time and that the sound of the instruments follows a stationary tone model aimed at woodwind, brass, and string instruments.

Since we perform *blind* separation, we do not make any prior assumptions specific to the sounds of the individual instruments, but we distinguish them based on the proximity to the entries of a *dictionary* which we learn in the process.

For the time-frequency representation of the audio signals, we use the sampled complex-valued output of the short-time Fourier transform, which can be interpreted as the analysis coefficients of a Gabor frame. This representation has the advantage of being perfectly linear and easy to project back to a time-domain signal, but it is not *pitch-invariant*; that is, the distance of the frequency axis corresponding to a certain musical interval varies based on the pitch of the tones.

The problem of identifying the pitch of the tones is non-convex on a global scale and possesses a large number of local minima. Therefore, general numerical optimization methods are not appropriate. Instead, we predict the parameters via a *U-Net* [1], which is a type of deep neural network. For the problematic parameters like pitch, we use policy gradients for training, which is a technique originating from deep reinforcement learning, cf. [2].

*Related Work*

The audio source separation problem can be formulated in a variety of settings. (See [3–5] for a thorough overview.) For the purpose of this work, we only regard the case that the input signal is single-channel and the separation is therefore always underdetermined. Different algorithms can be used if multiple channels are available (typically corresponding to microphones simultaneously recording the audio scene). Moreover, we always assume melodic instruments rather than speech or percussive instruments. While different kinds of prior information can be considered (such as specific characteristics of the sounds of the instruments, training data, or the musical score), we concentrate on the blind case with no prior information and instead rely on a learned parametric model for the sounds of the individual instruments.

Many algorithms for this problem are based on non-negative matrix factorization (NMF) of the spectrogram. In the simplest form, each tone of an instrument at a particular pitch has its own representation as a dictionary atom [6,7]. To make the representation of the sound of a particular instrument applicable at arbitrary pitch (*pitch-invariance*), one often employs *tensor factorization*, cf. [5]. In this case, the use of a log-frequency spectrogram (such as the constant-Q transform or the mel spectrogram) can be helpful, since it is also pitch-invariant in the sense that changing the fundamental frequency of a tone merely causes a shift in the representation [8–11].

A separation approach that is mathematically equivalent to NMF is *probabilistic latent component analysis* (PLCA) [12,13] which also exists in variants that use a pitch-invariant model on top of a log-frequency spectrogram [14,15]. The next step in abstraction is to model the individual harmonics separately while enforcing sparsity in the spectral representation [16].

Decreasing the variability, e.g., the number of parameters, in a model is generally beneficial since it reduces the risk of overfitting. However, we can go one step further by employing an explicit physical model for the tones of the instruments involving a minimum number of parameters. This has the additional advantage that while the previously discussed approaches with a log-frequency spectrogram can only be pitch-invariant if the instruments are tuned to the same log-frequency scale (*equal-temperament tuning*, cf. [17], defined as the frequency ratio corresponding to a musical interval being constant regardless of pitch), a physical model can be evaluated at any fundamental frequencies and sampled arbitrarily. This is crucial when dealing with acoustic instruments that either deliberately deviate from equal temperament or might simply be slightly out of tune.

With a continuous model, it is thus possible to use a pitch-invariant representation on a linear-frequency spectrogram. However, the challenge then is to identify the fundamental frequencies on a continuous domain. Duan et al. [18] use a peak detection and clustering algorithm to reduce the problem to a combinatorial one that can be approached via appropriate heuristics. Hennequin et al. [19], in a polyphonic single-instrument setting, consider the fundamental frequency as an optimizable parameter and use an NMF-type update rule, but they remark that this is only possible on a local scale due to the high number of local minima.

Even with a physical model, it turns out that the log-frequency spectrogram can still be helpful: Schulze and King [20] use its pitch-invariance property to obtain the approximate fundamental frequency of a tone via simple cross-correlation. After that, numerical optimization is performed to improve the estimate on a local scale. The optimization procedure also incorporates inharmonicity, which violates pitch-invariance, and variable width of the peaks in the spectrum which can occur, for instance, at tone boundaries. However, as explained there, using a log-frequency spectrogram inevitably results in a loss in frequency resolution which in this case needs to be mitigated via heavy preprocessing. Moreover, phase information is lost completely. Therefore, if given the choice, we argue that a linear-frequency representation should be preferred.

Due to the general success of deep neural networks, it is not surprising that they have also been applied to audio source separation problems. In fact, when it comes to supervised

separation (with labeled training data), they dominate the state of the art [21–25]. While supervised training is the "classical" way in which neural networks are used, it was demonstrated by Ulyanov el al. with the *deep image prior* (DIP) approach [26] that the structure of (convolutional) neural networks is inherently useful for representing natural images. This technique was used for image decomposition by Gandelsman et al. via the *double-DIP* algorithm [27]. Given this success, it was natural to also apply this method to audio data, leading to the *deep audio prior* approach by Tian et al. [28]. Based on this, Narayanaswamy et al. [29] used *generative adversarial networks* (GANs) trained on unlabeled training data as priors, further improving the quality of the output signals.

The problem with all the previously presented separation algorithms based on unsupervised training of neural networks is that they make the assumption that the separated signals are stochastically independent. This case is often referred to as the *cocktail party problem*, but it is different from polyphonic music, in which the tones are usually both rhythmically and harmonically aligned. Therefore, rather than relying on general stastistical properties of the signals, we use deep neural networks in conjunction with a parametric model.

*Policy gradients* were pioneered within reinforcement learning with the *REINFORCE* algorithm [30], which is designed to train a neural network to predict a discrete variable. While reinforcement learning has progressed towards *actor-critic* methods, cf. [2], and, famously, the *AlphaGo Zero* [31], *AlphaZero* [32], and *MuZero* [33] algorithms based on *Monte Carlo tree search* (MCTS), we stay relatively close to the original approach, but we extend the formulation by adding deterministic values, combining policy gradients with backpropagation gradients.

## 2. Data Model

### 2.1. Tone Model

An idealized model for the tones of woodwind, brass, and string instruments is that of the wave equation, which is a hyperbolic second-order partial differential equation. However, for certain string instruments, the stiffness in the strings is non-negligible, and this leads to the introduction of fourth-order terms which cause inharmonicity, cf. [34]. The corresponding solution consists of real-valued sinusoids, which consequently also appear in the audio signal. However, due to

$$\sin(\xi) = \frac{e^{i\xi} - e^{-i\xi}}{2i}, \qquad \cos(\xi) = \frac{e^{i\xi} + e^{-i\xi}}{2}, \tag{1}$$

we can also express them as complex exponentials. Since we do not need the negative exponential, our model of a tone of a musical instrument is as follows:

$$x(t) = \sum_h a_h \, e^{i2\pi f_h t}, \tag{2}$$

with

$$f_h = f_1^\circ h \sqrt{1 + bh^2}, \tag{3}$$

where $h = 1, \ldots, N_{\text{har}}$ are the *harmonics* (with $N_{\text{har}} \in \mathbb{N}$), $a_h \in \mathbb{C}$ is the complex *amplitude*, $f_h$ is the *frequency* for the specific harmonic, $f_1^\circ > 0$ is the *fundamental frequency* of the tone, and $b \geq 0$ is the *inharmonicity*.

For illustration, an artificial application of the tone model is provided in Figure 1. While the inharmonicity is exaggerated in comparison to real acoustic pianos, the increase in distance between the harmonics in the frequency domain is clearly visible. In the time domain, this has the effect that the overall signal is no longer periodic, even though the signals stemming from the individual harmonics are.

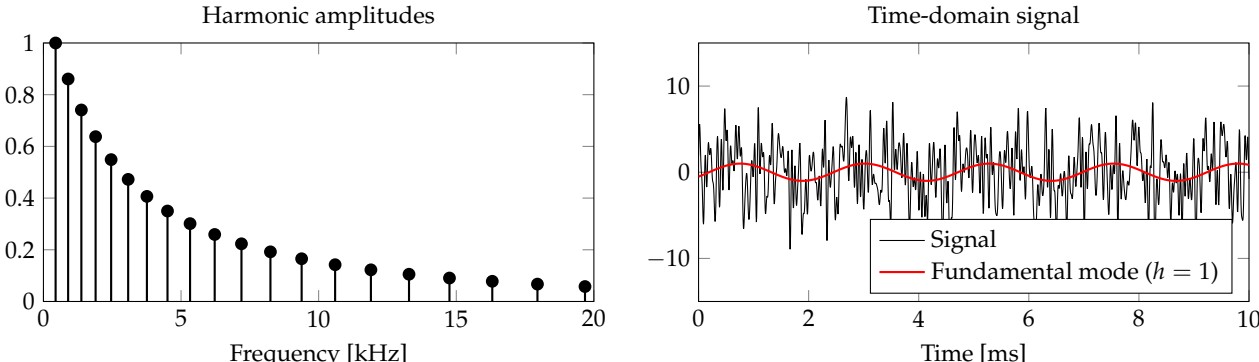

**Figure 1.** Illustrative example for the signal model with a fundamental frequency of $f_1^\circ = 440\,\text{Hz}$ and an inharmonicity parameter of $b = 10^{-2}$.

### 2.2. Time-Frequency Representation

Due to the sinusoidal nature of the tone model (2), it is advantageous to consider the signal in the frequency domain rather than the time domain. However, in reality, music is not stationary over longer periods of time; tones start, end, and change in volume or frequency. Therefore, we only consider the frequency spectrum of short excerpts in time, which is called a *time-frequency representation*. We compute the *short-time Fourier transform* (STFT) of a signal $X \in L_2(\mathbb{R})$ via:

$$\mathcal{V}_w X(t, f) = \int_{-\infty}^{\infty} X(\tau)\, w(\tau - t)\, e^{-i2\pi f \tau}\, \mathrm{d}\tau, \tag{4}$$

where $t, f \in \mathbb{R}$ correspond to the time and frequency axes and $w \in L_2(\mathbb{R})$ is the real-valued *analysis window*, cf. [35]. In our notation, the uppercase letters $X, Z, Y$ always refer to measured data, while their lowercase counterparts $x, z, y$ are our corresponding models. We sample the representation as:

$$Z[k, l] := \mathcal{V}_w X(\alpha k, \beta l)\, e^{i2\pi\alpha k \beta l}, \qquad k, l \in \mathbb{Z}, \tag{5}$$

with the time and frequency constants $\alpha, \beta > 0$. For $\alpha\beta < 1$ and with the Gaussian window

$$w(t) = \frac{1}{\sqrt{2\pi\zeta^2}} \exp\left(-\frac{t^2}{2\zeta^2}\right), \qquad \zeta > 0, \tag{6}$$

a so-called *Gabor frame* is formed, meaning that any function $X \in L_2(\mathbb{R})$ is uniquely determined by its respective set of *analysis coefficients* $Z[k, l]$, $k, l \in \mathbb{Z}$, cf. [35]. In practice, we only consider a finite number of indices $k = 1, \ldots, n_{\text{len}}$ (for time) and $l = 0, \ldots, n_{\text{spc}} - 1$ (for frequency) with $n_{\text{len}}, n_{\text{spc}} \in \mathbb{N}$, where we again neglect negative frequencies. If $X$ is real-valued, we have $Z[k, -l] = \overline{Z[k, l]}$.

For the tone model (2), the STFT yields:

$$\mathcal{V}_w x(t, f) = \sum_h a_h \exp\left(-\frac{(f - f_h)^2}{2\sigma^2} - i2\pi(f - f_h)t\right), \tag{7}$$

with $\zeta\sigma = 1/(2\pi)$. For tones at very low frequencies, the negative frequencies that were omitted from the tone model (2) can cause interference with the positive part of the frequency spectrum. With typical audio signals, this interference is not strong enough to become a problem. It would be straight-forward to add them back in, but this would increase the computational cost. When sampling according to (5), we obtain:

$$
\begin{aligned}
z[k,l] &= \mathcal{V}_w x(\alpha k, \beta l)\, e^{i2\pi\alpha k\beta l} \\
&= \sum_h a_h \exp\left(-\frac{(\beta l - f_h)^2}{2\sigma^2} - i2\pi(\beta l - f_h)\alpha k\right) e^{i2\pi\alpha k\beta l} \\
&= \sum_h a_h \exp\left(-\frac{(\beta l - f_h)^2}{2\sigma^2} + i2\pi f_h \alpha k\right).
\end{aligned}
\tag{8}
$$

Thus, for each harmonic $h$, the phase is constant for a fixed time index $k$.

Our default choice, assuming a sampling frequency of $f_s = 48\,\text{kHz}$, is:

$$
\zeta = \frac{1024}{f_s} = 21.\overline{3}\,\text{ms}, \quad \alpha = \frac{\zeta}{2} = \frac{512}{f_s} = 10.\overline{6}\,\text{ms}, \quad \beta = \frac{1}{12\zeta} = \frac{f_s}{122{,}88} = 3.906\,25\,\text{Hz}. \tag{9}
$$

The value of $\zeta$ is short enough to capture rhythm, while the quantity $1/(2\pi\zeta) \approx 7.46\,\text{Hz}$, that will become important later, is well below the fundamental frequencies considered. Our choice of $\alpha$ ensures that Gaussian windows spaced by $\alpha$ overlap narrowly.

For practical computations, it is common to limit the support of the window $w$ to $[-1/(2\beta), 1/(2\beta)]$. Due to $1/(2\beta) = 6\zeta$, our value for $\beta$ makes the resulting error negligible.

### 2.3. Dictionary Representation

In order to differentiate between instruments in a music recording, we make the simplifying assumption that the tones for each instrument $\eta = 1, \ldots, N_{\text{ins}}$ (where $N_{\text{ins}}$ is the total number of instruments) follow a characteristic pattern, namely that we can express the amplitudes of the harmonics as:

$$
a_h = a\, D[h, \eta]\, e^{i\tilde{\varphi}_h}, \qquad h = 1, \ldots, N_{\text{har}}, \tag{10}
$$

where $a \geq 0$ is the *global amplitude* of the tone, $D \in [0,1]^{N_{\text{har}} \times N_{\text{ins}}}$ is the *dictionary* containing the *relative amplitudes* for the harmonics, and $\tilde{\varphi}_h \in [-\pi, \pi)$ is the phase angle for the respective harmonic.

In a realistic music recording, the tones of the different instruments overlap and their parameters change over time. Thus, to construct an appropriate time-frequency model, we must equip the parameters with indices relating to the tones $j = 1, \ldots, m$ (where $m \in \mathbb{N}$ is the total number of simultaneously played tones) and the time frames $k = 1, \ldots, n_{\text{len}}$. We define the tone-wise and global model spectrograms, respectively, as:

$$
\begin{aligned}
z_j[k,l] &= \sum_h a_{j,h,k} \exp\left(-\frac{(\beta l - f_{j,h,k})^2}{2\sigma_{j,k}^2} + i2\pi f_{j,h,k}\alpha k\right) \\
&= \sum_h a_{j,k} D[h, \eta_{j,k}] \exp\left(-\frac{(\beta l - f_{j,h,k})^2}{2\sigma_{j,k}^2} + i\varphi_{j,h,k}\right)
\end{aligned}
\tag{11a}
$$

incorporating the phase shift via $\varphi_{j,h,k} := \tilde{\varphi}_{j,h,k} + 2\pi f_{j,h,k}\alpha k$, and:

$$
z[k,l] = \sum_j z_j[k,l]. \tag{11b}
$$

For signals consisting only of sinusoids, the spectrum is modeled precisely by (8), where the standard deviation $\sigma_{j,k}$ of the Gaussians is given as $\sigma_{j,k} = 1/(2\pi\zeta)$. However, especially at the beginning and end of a tone, boundary effects can occur, leading to the spectrum being better approximated by different values for $\sigma_{j,k}$. Therefore we include those as free parameters.

## 3. Learned Separation

### 3.1. Distance Function

The parametrized model $z[k, l]$ from (11) should match the time-frequency representation $Z[k, l]$ from (5) as closely as possible. To formalize this, we need to define a distance function. While the $\ell_2$ distance would be simple and "canonical," the problem is that it overemphasizes the correctness of the high-volume parts of the representation rather than the structural similarity. A common alternative is the $\beta$-divergence, cf. [36] (which is a generalization of the $\ell_2$ distance, the Kullback-Leibler divergence, and the Itakura-Saito divergence), but this leads to problems with unexplained noise in the spectrum. Instead, we use the distance measure introduced in [20]; for a given time frame $k$, we set $y = z[k, \cdot]$ and $Y = Z[k, \cdot]$ and define:

$$d_{2,\delta}^{q,\text{abs}}(Y, y) = \frac{1}{2} \sum_l \Big( \big(|Y[l]| + \delta\big)^q - \big(|y[l]| + \delta\big)^q \Big)^2, \qquad q \in (0, 1], \quad \delta > 0. \tag{12}$$

The $q$ exponent has the purpose of *lifting* the low-volume parts of the representation in order to increase their relevance. The canonical choice is $q = 1/2$, since this is the lowest value to keep the expression convex in $y$. The value for $\delta$ can be low, as it is merely there to ensure differentiability at $y[l] = 0$. Unlike the $\beta$-divergence, the distance function in (12) is symmetric, but it is still not a metric in the mathematical sense.

### 3.2. Model Fitting

Even though (12) is convex in $y$ for $q = 1/2$, the spectrum $y$ itself is not point-wise globally convex in all the parameters appearing in (11). Therefore, conventional optimization methods based on gradient descent are not a good choice for minimizing $d_{2,\delta}^{q,\text{abs}}(Y, y)$. Instead, we use a deep neural network to predict the tone parameters. While some of these parameters (the *deterministic parameters*) can be trained normally via backpropagation, we treat the "problematic" parameters as *stochastic parameters* which are trained via *policy gradients* [30].

The neural network is applied in such the way that it predicts the parameters for one tone at a time. After each prediction, the spectrum for that tone is computed and provided back to the network for the following steps.

#### 3.2.1. Parameter Representation

We first have to decide which parameters are stochastic and which ones are deterministic. The fundamental frequency parameter $f_{j,1,k}^{\circ}$ is clearly one of those in which $y$ is not convex; however, it does have a useful gradient on a local scale. We therefore split this parameter into two parameters as $f_{j,1,k}^{\circ} = \beta(\nu_{j,k} + \tilde{\nu}_{j,k})$, where the values for $\nu_{j,k} \in \mathbb{N}$ are discrete and those for $\tilde{\nu}_{j,k} \in \mathbb{R}$ are continuous. We treat $\nu_{j,k}$ as a stochastic parameter and $\tilde{\nu}_{j,k}$ as a deterministic one.

If we limit each instrument to exactly one tone (therefore, $m = N_{\text{ins}}$), the instrument parameter $\eta_{j,k}$ is not required mathematically. However, it makes sense to include it from a practical algorithmic perspective since this allows for a network architecture that sequentially extracts one tone after another while freely choosing the extraction order (see Section 3.2.2). As $\eta_{j,k}$ is discrete, it is impossible to obtain a gradient. Thus, we model it as a stochastic parameter following a categorical distribution.

For $\sigma_{j,k}$, while it is possible to find examples in which the distance $d_{2,\delta}^{q,\text{abs}}(Y, y)$ is not convex with respect to it, the gradient around the theoretical value is usually good, so we can treat it as a deterministic parameter.

The inharmonicity parameter $b_{j,k}$ is also continuous and has a good gradient around the optimum, but depending on the characteristics of the instrument, there can exist local optima. Therefore we do not rely on backpropagation and instead treat it as a stochastic parameter following a gamma distribution (see Figure 2). The reason why we chose a gamma distribution is that it is non-negative and it can have two qualitatively different shapes:

Either it tends towards $\infty$ at zero (which is useful to model tones without inharmonicity), or it is bell-shaped around a finite maximum (to model tones with inharmonicity). Approaching infinity, it always decays exponentially, and in the edge case between the two shapes, it matches an exponential distribution.

In the tone amplitudes $a_{j,k}$, the problem is convex, and therefore they are treated as deterministic parameters. In total, the stochastic parameters for each tone are $\varpi_{s,j,k} = (\nu_{j,k}, \eta_{j,k}, b_{j,k})$, and the deterministic parameters are $\varpi_{d,j,k} = (a_{j,k}, \tilde{\nu}_{j,k}, \sigma_{j,k})$. When it is clear which time frame we are considering, we can drop the dependency on $k$ and summarize:

$$\varpi_s = (\varpi_{s,1}, \ldots, \varpi_{s,m}) = (\varpi_{s,1,k}, \ldots, \varpi_{s,m,k}), \tag{13a}$$

$$\varpi_d = (\varpi_{d,1}, \ldots, \varpi_{d,m}) = (\varpi_{d,1,k}, \ldots, \varpi_{d,m,k}). \tag{13b}$$

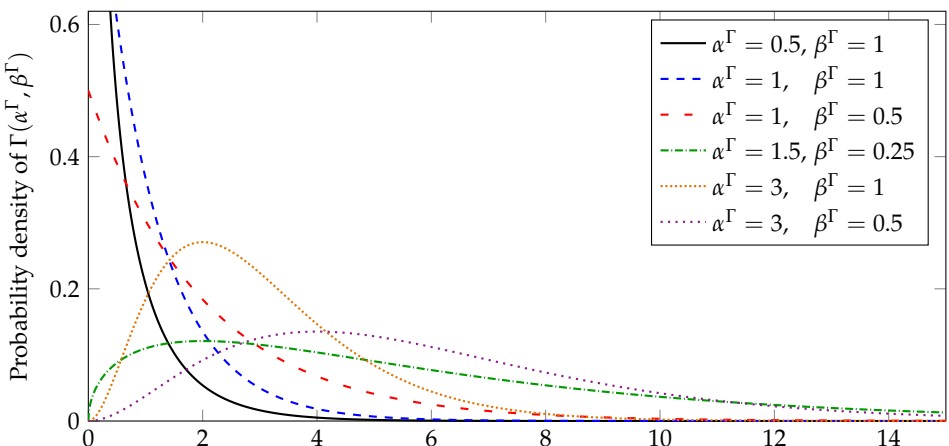

**Figure 2.** Probability density functions of gamma distributions for different parameter choices. For $\alpha^\Gamma \geq 1$, the function has the mode at $(\alpha^\Gamma - 1)/\beta^\Gamma$, while for $\alpha^\Gamma < 1$, the function tends to $\infty$ at zero. In our proposed method, the network selects a distribution shape for each inharmonicity coefficient $b_j$ by its outputs $\alpha_j^\Gamma, \beta_j^\Gamma$.

### 3.2.2. Policy Gradients

We use a neural network with network parameters $\theta$ both to give the *policy* $\pi_\theta(\varpi_s|Y)$, which is the (discrete or continuous) probability density for the stochastic parameters $\varpi_s$ given the input spectrum $Y$ and also to compute the deterministic parameters $\varpi_d$ from the input and the stochastic parameters. With this, we can express the *loss function* as:

$$L(\varpi_s, \theta, D, Y) := L\big(\varpi_s, \varpi_d(\varpi_s, \theta, Y), D, Y\big) = d_{2,\delta}^{q,\text{abs}}\big(Y, y(\varpi_s, \varpi_d, D)\big) = d_{2,\delta}^{q,\text{abs}}(Y, y). \tag{14}$$

For the expected loss, we follow the usual computation, cf. [2], but also apply the product rule for the deterministic parameters:

$$\begin{aligned}
&\nabla_\theta \, \mathrm{E}_{\pi_\theta(\varpi_s|Y)}\big[L(\varpi_s, \theta, D, Y)\big] \\
&= \nabla_\theta \int \pi_\theta(\varpi_s|Y) \, L(\varpi_s, \theta, D, Y) \, d\varpi_s \\
&= \int \nabla_\theta \pi_\theta(\varpi_s|Y) \, L(\varpi_s, \theta, D, Y) + \pi_\theta(\varpi_s|Y) \, \nabla_\theta L(\varpi_s, \theta, D, Y) \, d\varpi_s \\
&= \int \pi_\theta(\varpi_s|Y) \left( \frac{\nabla_\theta \pi_\theta(\varpi_s|Y)}{\pi_\theta(\varpi_s|Y)} \, L(\varpi_s, \theta, D, Y) + \nabla_\theta L(\varpi_s, \theta, D, Y) \right) d\varpi_s \\
&= \mathrm{E}_{\pi_\theta(\varpi_s|Y)}\big[\nabla_\theta \log \pi_\theta(\varpi_s|Y) \, L(\varpi_s, \theta, D, Y) + \nabla_\theta L(\varpi_s, \theta, D, Y)\big],
\end{aligned} \tag{15}$$

where we refer to the first term in the sum as the *policy gradient* and on the second term as the *backpropagation gradient*. The applicability of the Leibniz integral rule can be shown under realistic conditions.

The total dimensionality of $\omega_s$ is too high to represent $\pi_\theta(\omega_s|Y)$ as a whole. Therefore, we decompose the log-probability density into:

$$
\begin{aligned}
&\log \pi_\theta(\omega_s|Y) \\
&= \log\big(\pi_\theta(\omega_{s,1}|Y) \cdot \pi_\theta(\omega_{s,2}|Y,\omega_{s,1}) \cdots \pi_\theta(\omega_{s,m}|Y,\omega_{s,1},\ldots,\omega_{s,m-1})\big) \\
&= \log \pi_\theta(\omega_{s,1}|Y) + \log \pi_\theta(\omega_{s,2}|Y,\omega_{s,1}) + \ldots + \log \pi_\theta(\omega_{s,m}|Y,\omega_{s,1},\ldots,\omega_{s,m-1}).
\end{aligned}
\tag{16}
$$

In practice, we first sample the stochastic parameters $\omega_{s,1}$ for the first tone, then obtain the deterministic parameters $\omega_{d,1}$, and from these compute a model spectrum $y_1$. For each additional tone $j = 2,\ldots,m$, we gain $\omega_{s,j}$ and $\omega_{d,j}$ depending on the previous spectra $y_1,\ldots,y_{j-1}$, which encapsulate all the relevant information about the parameters of the previous tones.

The parameters $\nu_j, \eta_j$ are sampled via a joint categorical distribution, making sure that no instrument plays more than one tone. For each possible value of $(\nu_j, \eta_j)$, the network gives a value for the deterministic parameters $\omega_{d,j}$ as well as for the parameters $\alpha_j^\Gamma, \beta_j^\Gamma$ for the gamma distribution for $b_j$.

### 3.3. Phase Prediction

Of the tone parameters for (10), we still need the phase angles $\varphi_{j,h}$. Canonically, we could represent them as a vector that is output by the network for each possible choice of $\nu_j, \eta_j$, but this would lead to high dimensionality. Instead, we let the network emit a single artificial spectrum $v_j \in \mathbb{C}^{n_{spc}}$ for each possible instrument choice $\eta_j$. This is used as the right-hand side of a least-squares problem for determining the coefficients $c_{j,h} \in \mathbb{C}$:

$$
\min_{(c_{j,h})} \frac{1}{2} \sum_l \left| \sum_h c_{j,h} \cdot \exp\left( -\frac{\left(\beta l - f_{j,1}^\circ h\sqrt{1+b_j h^2}\right)^2}{2\sigma_j^2} \right) - v_j[l] \right|^2, \qquad l = 0,\ldots,n_{spc}-1, \tag{17}
$$

from which we extract the phase angles as $\varphi_{j,h} = \arg c_{j,h}$. We apply some $\ell_2$ regularization to improve the condition number of the system. For $c_{j,h} = 0$, the phase would be ill-defined, but since the magnitudes of these coefficients are stabilized via the objectives introduced in Section 3.4, we do not realistically expect this case to happen.

With this approach, the frequency dimension of the network output is used differently than for the other tone parameters. Instead of providing an output for each possible tone choice $\nu_j$, the network computes a single spectrum, which then determines the phases of all harmonics. We expect the spatial structure of the U-Net architecture with respect to the frequency dimension to be beneficial for this task. While $c_{j,h}$ is not explicitly given as a conditional value depending on $\omega_s$, the computation of $c_{j,h}$ in (17) does depend on all the stochastic parameters (even $b_j$), and since typically $n_{spc} > N_{har}$, the network has some freedom to output $v_j$ such that $c_{j,h}$ take different values depending on the other parameters.

Since $v_j$ is the right-hand side of a linear least-squares system, optimizing $y$ with respect to $v_j$ is convex. Therefore, training is done deterministically via backpropagation through the pseudo-inverse. Moreover, we include the additional gradient that occurs with respect to the left-hand-side parameters that appear inside the exponential. Computation of the gradient of the solution of a least-squares system with respect to the left-hand side is usually included in automatic differentiation frameworks, but to obtain it explicitly, one can repeatedly apply the Woodbury formula.

In the case where the positions of the peaks from the model perfectly match those from the spectrum $Y$ without any overlap or additive noise, the choice $v_j = Y$ gives the ideal phase values $\varphi_{j,h}$ for all $j = 1,\ldots,m$ and $h = 1,\ldots,N_{har}$. While this exact case is not realistic (not least since Gaussians have unbounded support), a network with skip connections can quickly learn to predict good approximate phase values.

### 3.4. Complex Objectives

So far, we have only used the parameters $c_{j,h}$ from (17) in order to determine the phase values $\varphi_{j,h}$. However, we can also use them for a different purpose: While the dictionary representation (10) is necessary in order to distinguish the instruments, it is never fully accurate since the relation of the amplitudes of the harmonics can vary slightly even for the same instrument. The typical remedy for this is *spectral masking*, but as explained in [20], this process does not properly deal with interference. Instead, we create a *direct prediction*, in which we replace the tone amplitudes and dictionary entries in (11) with the parameters $c_{j,h,k}$ (with time-frame dependency added back in):

$$z_j^{\text{dir}}[k,l] = \sum_h c_{j,h,k} \exp\left(-\frac{(\beta l - f_{j,h,k})^2}{2\sigma_{j,k}^2}\right), \tag{18a}$$

$$z^{\text{dir}}[k,l] = \sum_j z_j^{\text{dir}}[k,l]. \tag{18b}$$

For a fixed $k$, we set $y^{\text{dir}} = z^{\text{dir}}[k,\cdot]$.

The distance function from (12) only considers the absolute value and ignores the phase entirely. This is not necessarily a problem, but knowing the exact phase angles would be useful for resynthesis, and it makes more sense as a training objective. Therefore, we introduce a modified distance function that respects the phase:

$$d_{2,\delta}^{q,\text{rad}}(Y,y) = \frac{1}{2}\sum_l \left|(|Y[l]| + \delta)^q \cdot \frac{Y[l]}{|Y[l]|} - (|y[l]| + \delta)^q \cdot \frac{y[l]}{|y[l]|}\right|^2. \tag{19}$$

To avoid division by zero, we add a tiny positive constant to the denominators.

We can now compare $y$ to $Y$ (as before), $y^{\text{dir}}$ to $Y$, and $y$ to $y^{\text{dir}}$, and we can choose between the distance functions $d_{2,\delta}^{q,\text{abs}}$ and $d_{2,\delta}^{q,\text{rad}}$ as defined in (12) and (19). While it would be ideal to have the phases matching in all spectra, there are some caveats:

- It takes a number of training iterations for $v_j$ to give a useful value. In the meantime, the training of the other parameters can go in a bad direction.
- If the discrepancy between $y$ and $y^{\text{dir}}$ is high *and* there is a lot of overlap between the peaks (typically from different tones), the optimal phase values for $y$ and $y^{\text{dir}}$ may be significantly different. An example for this is displayed in Figure 3: The two peaks (red and blue) each have different phases, but by design, those are identical between the predictions. However, since the dictionary prediction is less flexible, its amplitude magnitudes of the harmonics often do not accurately match the input spectrum $Y$, which shifts the phase in the overlapping region. Thus, attempting to minimize both $d_{2,\delta}^{q,\text{rad}}(Y,y^{\text{dir}})$ and $d_{2,\delta}^{q,\text{rad}}(Y,y)$ would lead to a conflict regarding the choice of common phase values.

Therefore, we continue to compare $y$ and $Y$ via $d_{2,\delta}^{q,\text{abs}}$ (without the phase), and we use $d_{2,\delta}^{q,\text{rad}}$ for the comparison between $y^{\text{dir}}$ and $Y$, since $y^{\text{dir}}$ is the spectrum that we will end up using for resynthesis, so we aim for the phase to be correct. Between $y$ and $y^{\text{dir}}$, we can compare tone-wise, but since some discrepancy is to be expected, we only associate it with a small penalty; the purpose is to regularize $y^{\text{dir}}$ for the case that the peaks of different tones overlap so much that the $c_{j,h}$ are not unique (see Figure 4).

For this task, we employ the loss terms $d_{2,\delta}^{q,\text{rad}}(y_j^{\text{dir}},y_j)$ to compare the tone spectra $y_j^{\text{dir}} := z_j^{\text{dir}}[k,\cdot]$ and $y_j := z_j[k,\cdot]$ in both magnitude and phase. However, since the individual peaks making up $y_j^{\text{dir}}$ and $y_j$ necessarily have the same phases, the difference between $d_{2,\delta}^{q,\text{rad}}$ and $d_{2,\delta}^{q,\text{abs}}$ only matters if there is significant overlap between the peaks within the same tone, which is the case at very low fundamental frequencies.

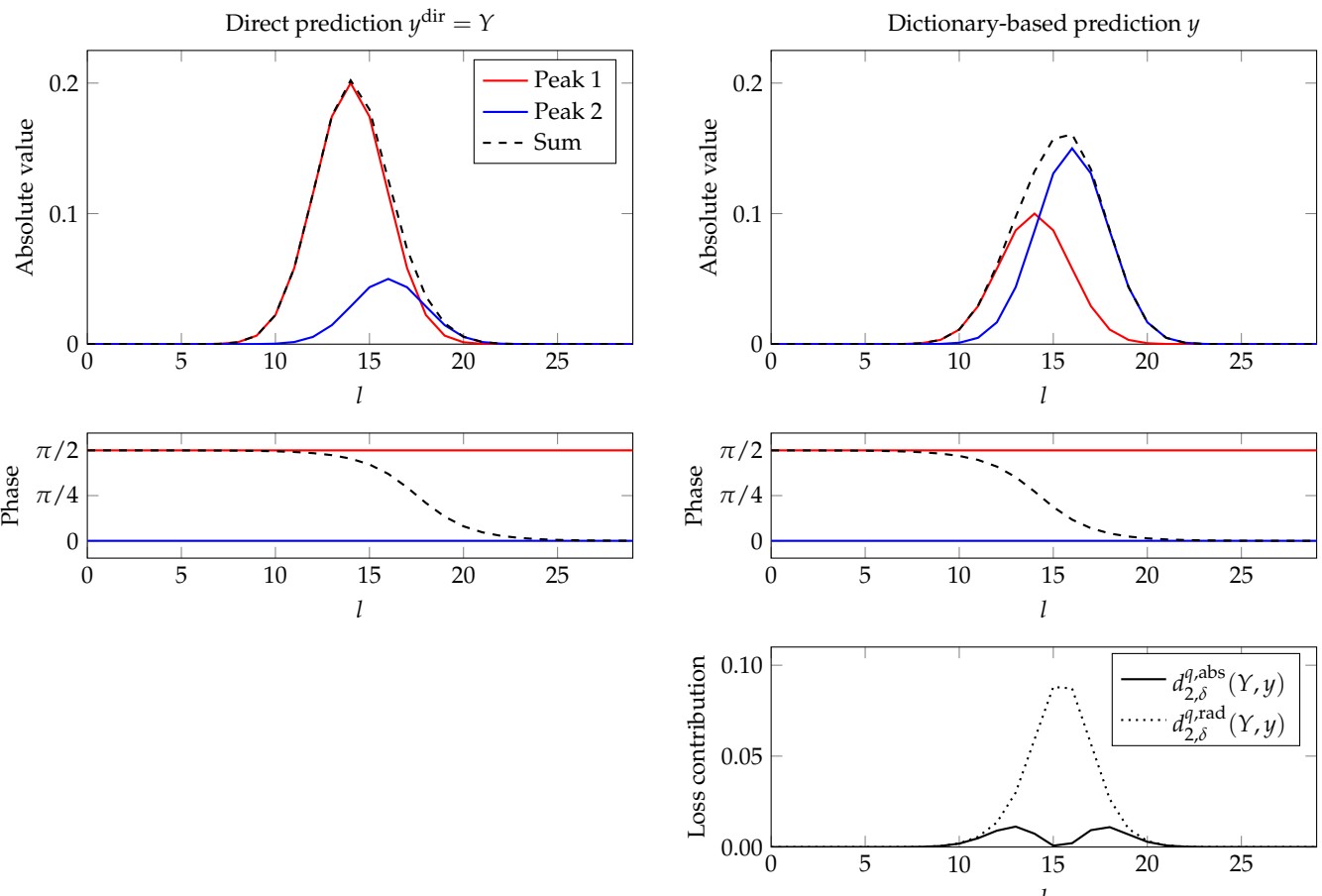

**Figure 3.** Example showing the interference of two overlapping peaks. Each peak models the contribution of one harmonic of a tone to the spectrum (cf. (8)). The left plots show a part of a direct prediction $y^{\mathrm{dir}}$, which is assumed to equal the true spectrum $Y$ for this example, and the right plots show a dictionary-based prediction $y$ with deviating amplitudes. Due to the different amplitudes, also the phases mix differently, leading to a high value of $d_{2,\delta}^{q,\mathrm{rad}}(Y,y)$. The phases could be optimized for $y$ (by increasing the phase for peak 1 and/or peak 2), but this would lead to suboptimal phases in $y^{\mathrm{dir}}$. In contrast, the used loss $d_{2,\delta}^{q,\mathrm{abs}}(Y,y)$ does not depend on the phase.

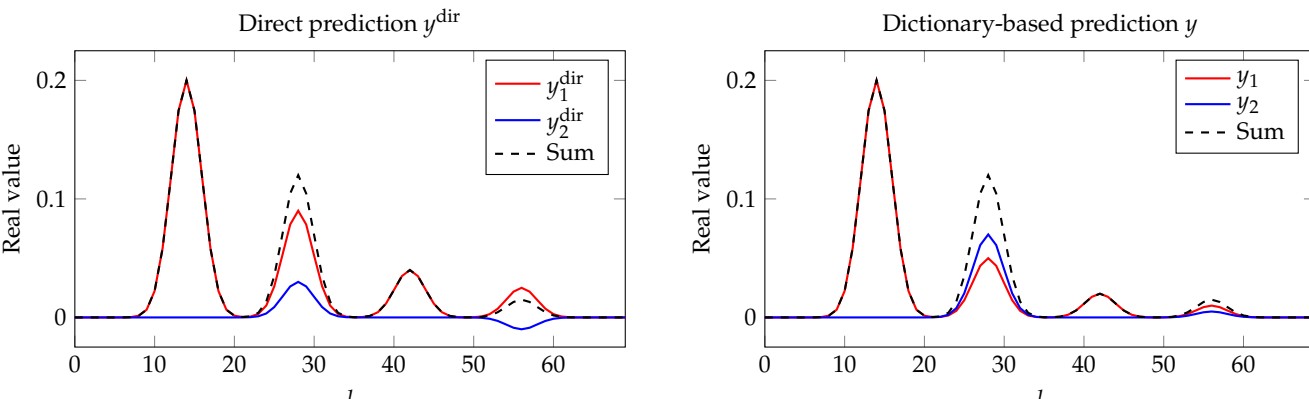

**Figure 4.** Example showing non-uniqueness of the tone separation in the direct prediction. In the direct prediction (**left**) the separation of the two instruments is different from the dictionary-based prediction $y$ (**right**). For this example, we assume the dictionary-based separation to be correct, so ideally the direct separation would be the same. However, the total spectrum (Sum) of the incorrect separation equals the true total spectrum and thus also achieves the optimal loss value $d_{2,\delta}^{q,\mathrm{rad}}(Y,y^{\mathrm{dir}})$. This motivates to regularize the individual tones $y_j^{\mathrm{dir}}$ of the direct prediction using $y_j$. The imaginary parts of the spectra are assumed to be all zero for this example.

The loss functions $d_{2,\delta}^{q,\mathrm{rad}}$ and $d_{2,\delta}^{q,\mathrm{abs}}$ are both based on the $\ell_2$ loss, so they do not induce sparsity. Thus, if there is linear dependency in the dictionary or, more likely, there is a discrepancy between the dictionary model for an instrument and an actual tone played by that instrument, then one single tone played by an instrument may get identified as multiple ones, either with the same fundamental frequency or with overlapping harmonics. Thus, we introduce an additional *sparsity parameter* $u_j \in \{0,1\}$ that indicates whether a certain tone is present at all, and we henceforth include it in the set of stochastic parameters $\varpi_{\mathrm{s},j}$. Whenever $u_j = 0$, we discard the tone in the *sparse prediction* given by

$$y^{\mathrm{spr}} = \sum_j u_j\, y_j \tag{20}$$

and discount part of the loss. In terms of architecture, the parameter $u_j$ is modeled as a Bernoulli distribution.

With $y_j^{\mathrm{dir}} = z_j^{\mathrm{dir}}[k,\cdot]$, $y_j = z_j[k,\cdot]$, and $y^{\mathrm{dir}} = z^{\mathrm{dir}}[k,\cdot]$ again, we define the loss as:

$$L(\varpi_{\mathrm{s}},\theta,D,Y) = \mu_1\, d_{2,\delta}^{q,\mathrm{abs}}(Y,y^{\mathrm{spr}}) \cdot \lambda^{\sum_j(1-u_j)} + \mu_2\, d_{2,\delta}^{q,\mathrm{rad}}(Y,y^{\mathrm{dir}}) + \frac{\mu_3}{m}\sum_j d_{2,\delta}^{q,\mathrm{rad}}(y_j^{\mathrm{dir}},y_j), \tag{21}$$

choosing $\lambda = 0.9$. This value is somewhat arbitrary, so we do not use it to enforce sparsity directly in $y$. Instead, we compute the loss in the first term based on $y^{\mathrm{spr}}$ so that the parameters are *compatible* with a sparse solution and do not rely on redundant tones. However, additional tones can still appear in $y^{\mathrm{dir}}$ to reduce the distance to $Y$ and, by extension, also in $y$ to reduce the distance to $y^{\mathrm{dir}}$.

We give both distances to $Y$ the same loss coefficients while the regularization is supposed to be small. In practice, we find that $\mu_1 = 10$, $\mu_2 = 10$, $\mu_3 = 1$ is a good choice.

### 3.5. Sampling for Gradient Estimation

In order to apply a gradient descent method, we need to compute the expectation in (15). However, the set of possible parameters $\varpi_{\mathrm{s}}$ is much too large to do so analytically, so we have to estimate it instead. For this, we use:

$$\hat{g}_{\pi_\theta,Y} = \frac{1}{S}\sum_{i=1}^{S}\Big(\nabla_\theta \log \pi_\theta(\varpi_{\mathrm{s}}^i|Y)\cdot\big(L(\varpi_{\mathrm{s}}^i,\theta,D,Y) - C(\theta,D,Y)\big) + \nabla_\theta L(\varpi_{\mathrm{s}}^i,\theta,D,Y)\Big), \tag{22}$$

with $\varpi_{\mathrm{s}}^1,\ldots,\varpi_{\mathrm{s}}^S \sim \pi_\theta(\varpi_{\mathrm{s}}|Y)$, where $S \in \mathbb{N}$ is the number of samples and $C(\theta,D,Y)$ is the *baseline*, cf. [2]. The baseline plays a crucial role in reducing the variance of the gradient, and a common choice is:

$$C(\theta,D,Y) = \mathrm{E}_{\pi_\theta(\varpi_{\mathrm{s}}|Y)}\big[L(\varpi_{\mathrm{s}},\theta,D,Y)\big]. \tag{23}$$

As long as the baseline is independent of the samples $\varpi_{\mathrm{s}}^1,\ldots,\varpi_{\mathrm{s}}^S$, the estimator (22) is unbiased. However, the easiest way to estimate the baseline is via:

$$\hat{C}(\theta,D,Y) = \frac{1}{S}\sum_{i=1}^{S}L(\varpi_{\mathrm{s}}^i,\theta,D,Y), \tag{24}$$

which obviously depends on the samples and therefore introduces a bias factor of $(S-1)/S$ for the policy gradient. If desired, one could correct for this bias by dividing by this factor, but it is generally advisable to be conservative about the policy gradient, so we choose to leave it uncorrected. However, the value for $S$ should be chosen large enough for (24) to be a good estimator.

During the training, it is expected that the policy will become increasingly deterministic, so less exploration will be performed. The problem with this is that multiple parameters need to be trained, and it is easy for the training process to get stuck in local

minima. Therefore, we should encourage the exploration of more parameter choices even when the policy has stabilized. Inspired by the algorithm from [31], we thus define:

$$\pi_\theta^{r_j}(\varpi_{s,j}|Y) := \frac{\pi_\theta(\varpi_{s,j}|Y)^{r_j}}{\int \pi_\theta(\varpi_{s,j}|Y)^{r_j} \, d\varpi_{s,j}}, \qquad r_j > 0. \tag{25}$$

The $r_j$ parameter can be chosen separately for each tone; with $R = (r_1, \ldots, r_m)$, we further define:

$$\pi_\theta^R(\varpi_s|Y) := \pi_\theta^{r_1}(\varpi_{s,1}|Y) \cdot \pi_\theta^{r_2}(\varpi_{s,2}|Y, \varpi_{s,1}) \cdots \pi_\theta^{r_m}(\varpi_{s,m}|Y, \varpi_{s,1}, \ldots, \varpi_{s,m-1}). \tag{26}$$

We set $S = 3^m$ and let $R_1, \ldots, R_S$ be all the combinations of $m$ elements out of the set $\{1, 0.1, 0.01\}$, which turn out to be reasonable magnitudes. Additionally, we do not only accept the bias which causes an underestimation of the policy gradient, but we also artificially scale it down by a factor of 10. Since the policy gradient and the backpropagation gradient model different parameters, this is conceptually not a problem; however, if the gradient is scaled *too* much, the interdependencies of the values in the network can then cause instabilities in the output of the stochastic parameters. Our modified gradient estimator is:

$$\hat{g}_{\theta,Y} = \frac{1}{S} \sum_{i=1}^{S} \left( \frac{1}{10} \nabla_\theta \log \pi_\theta^{R_i}(\varpi_s^i|Y) \cdot (L(\varpi_s^i, \theta, D, Y) - \hat{C}(\theta, D, Y)) + \nabla_\theta L(\varpi_s^i, \theta, D, Y) \right), \tag{27}$$

where each tone parameter set $\varpi_{s,j}^i$ is sampled according to the value of $r_j$ inside $R_i$. This includes the sparsity parameter $u_j$, but the inharmonicity $b_j$ is exempt from this modification altogether and always sampled according to $\pi_\theta$.

Since the way of sampling also affects the empirical baseline $\hat{C}(\theta, D, Y)$, it is no longer an estimator for $C(\theta, D, Y)$ but simply the mean loss among the samples. We observe this to be a good stabilizer for the gradient. Via our choice of $R_1, \ldots, R_S$ we make sure that one sample is drawn with $(r_1, \ldots, r_m) = (1, \ldots, 1)$ from the original distribution $\pi_\theta$. Moreover, even with increased exploration in some of the tones, there is always a combination with $r_j = 1$ for the other tones. Therefore, rather than destabilizing all tones at the same time, part of the exploration is only performed selectively on specific tones with the other ones still sampled via $\pi_\theta$.

We also need to train the dictionary $D$; since the representation of $D$ is not conditional but simply a variable of size $N_{\text{har}} \times N_{\text{ins}}$, exploration is not desired. In fact, even though the policy gradient on the dictionary exists for all tones but the first (via the dependency of $y_j$ on $D$), we ignore it to increase training stability. However, the variables $\varpi_s^i$ were sampled according to $\pi_\theta^{R_i}$ rather than $\pi_\theta$, so to get a stable estimate for $D$, we "undo" this modification by multiplying with the ratio between the probability densities. We estimate the gradient in $D$ via:

$$\hat{g}_{D,Y} = \frac{\sum_{i=1}^{S} \rho_i \, \nabla_D L(\varpi_s^i, \theta, D, Y)}{\sum_{i=1}^{S} \rho_i}, \qquad \rho_i = \frac{\pi_\theta(\varpi_s|Y)}{\pi_\theta^{R_i}(\varpi_s|Y)}, \tag{28}$$

where $\pi_\theta^{R_i}$ is defined according to (26). This method is called *weighted importance sampling*, cf. [2].

### 3.6. Network Architecture

We use a U-Net architecture with seven downsampling/upsampling steps; strides of $4, 4, 4, 4, 4, 3, 2$; and $80, 160, \ldots, 560$ one-dimensional filters with a size of 5. Finally, we add two more convolutional layers with 80 filters of sizes 3 and 1, respectively, before the linear output layer. All of the hidden layers have ReLU activation. The first hidden layer is a *CoordConv* layer [37], which means that it is supplied with a linear range from 0.01

to 0 as an additional input channel. These design parameters were obtained via manual experimentation.

For the first tone, the network is supplied with the input spectrum $Y$ and the absolute value spectrum $|Y|$. From this, the spectra $y_1$ and $y_1^{\mathrm{dir}}$ are computed. For each following tone $j = 2, \ldots, m$, the network receives the residuals $Y - y_1 - \ldots - y_{j-1}$ and $Y - y_1^{\mathrm{dir}} - \ldots - y_{j-1}^{\mathrm{dir}}$ along with their absolute values as well as the computed tone spectra $y_1, \ldots, y_{j-1}$ and $y_1^{\mathrm{dir}}, \ldots, y_{j-1}^{\mathrm{dir}}$. From these values, it then yields the spectra $y_j$ and $y_j^{\mathrm{dir}}$. For those input components of the network that do not yet receive a spectrum for a particular tone, a constant 0 vector is given as input to the network instead. The data flow is illustrated in Figure 5.

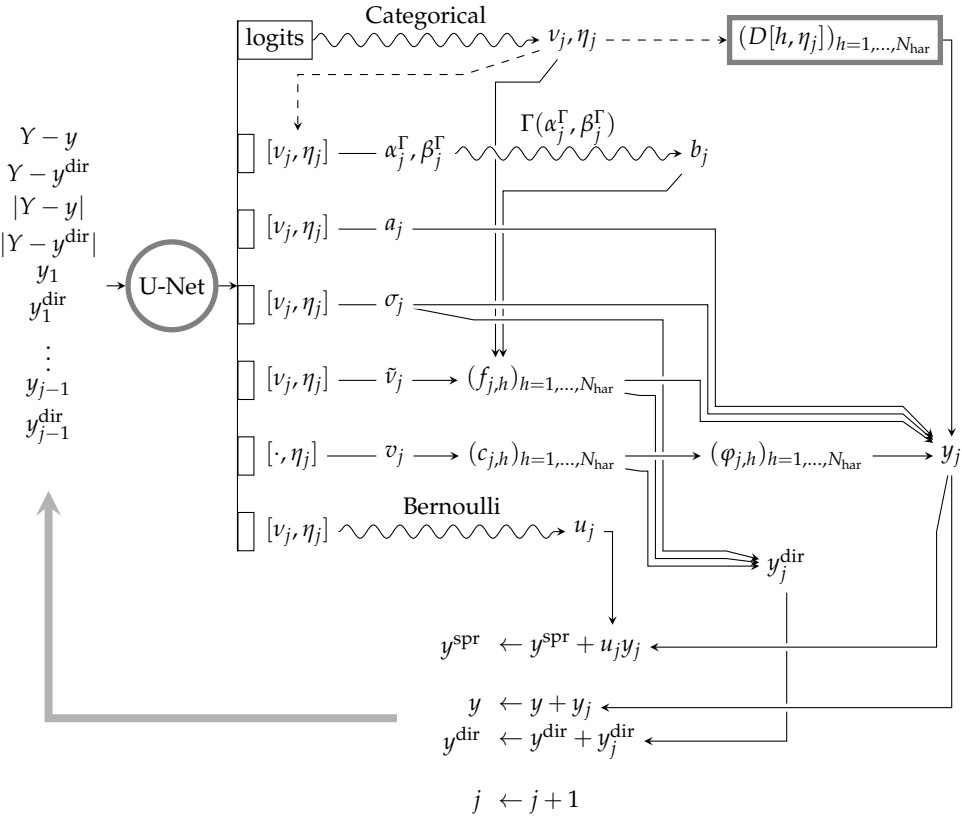

**Figure 5.** Sampling architecture.

Since the amplitudes $(a_j)$ are supposed to be non-negative, we apply the absolute value function to the respective output components of the network. The widths $(\sigma_j)$ are kept positive via softplus, and they are clipped such that the value does not get too close to 0. For the continuous frequency offsets $(\tilde{v}_j)$, a tanh function is used to keep them inside the interval $(-5, 5)$. The positive parameters $(\alpha_j^\Gamma, \beta_j^\Gamma)$ are obtained after applying the exponential function, and the probabilities for the Bernoulli distribution for the sparsity parameters $(u_j)$ are mapped into the interval $(0, 1)$ via a sigmoid function. The joint categorical distribution for the discrete frequencies $(v_j)$ and instrument indices $(\eta_j)$ is given in vectorial form as non-normalized log-probabilities, so we apply the *softmax* mapping in order to obtain a valid discrete distribution. Doing this, we have to make sure that each instrument can only play one tone at a time by excluding that instruments that have already been assigned a tone from the sampling. For each parameter output by the network, we add a trainable scaling layer.

In order to protect against potentially degenerate network output or gradients in the case of zeros in the input vector, a minimal amount of Gaussian noise is always added to $Y$ prior to prediction, on a level that is negligable for any normal audio signals.

The dictionary entries are supposed to be elements of the interval $[0,1]$. Non-negativity is usually satisfied automatically due to $a_j \geq 0$, and for the upper bound, we add the loss $\frac{1}{N_{\text{ins}}} \sum_\eta (\log(\max_h D[h,\eta]))^2$ to the training of the dictionary.

### 3.7. Training

The network weights are Glorot-initialized and the biases are initially set to 0. The initial values for the instruments in the dictionary are exponentially decaying sequences along the harmonics:

$$D_0[h,\eta] = \left(\frac{0.5}{\eta}\right)^{h-1}. \tag{29}$$

We partition the spectra $Z[k,\cdot]$, $k = 1,\ldots,n_{\text{len}}$, into random batches of size 6. We then train on each batch with the AdaMax algorithm [38]. For each epoch, new random batches are assigned. For the dictionary, we also use AdaMax, but with a reduced learning rate of $10^{-4}$. Moreover, analogously to [20], for the denominator in AdaMax, we consider the maximum over all the harmonics of the particular instrument when training $D$. The entire procedure is outlined in Algorithm 1.

---

**Algorithm 1** Training scheme for the network and the dictionary, based on AdaMax [38]. Upper bound regularization of $D$ and batch summation (see Sections 3.6 and 3.7) are not explicitly stated.

---

**Input:** $Z, \theta, D$
**Parameters:** $T \in \mathbb{N}$, $\kappa_\theta > 0$, $\kappa_D > 0$, $\beta_1 \in (0,1)$, $\beta_2 \in (0,1)$, $\varepsilon > 0$

    $\gamma_{\theta,1} \leftarrow 0$
    $\gamma_{\theta,2} \leftarrow 0$
    $\gamma_{D,1} \leftarrow 0$
    $\gamma_{D,2} \leftarrow 0$
    **for** $\tau = 1,\ldots,T$ **do**
        choose $Y$ out of $\{Z[k,\cdot] : k = 1,\ldots,n_{\text{len}}\}$
        $\gamma_{\theta,1} \leftarrow \beta_1 \gamma_{\theta,1} + (1-\beta_1)\hat{g}_{\theta,Y}$
        $\gamma_{\theta,2} \leftarrow \max(\beta_2 \gamma_{\theta,2}, |\hat{g}_{\theta,Y}|)$
        $\theta \leftarrow \theta - \frac{\kappa_\theta}{1-\beta_1^\tau} \cdot \frac{\gamma_{\theta,1}}{\gamma_{\theta,2}+\varepsilon}$
        $\gamma_{D,1} \leftarrow \beta_1 \gamma_{D,1} + (1-\beta_1)\hat{g}_{D,Y}$
        $\gamma_{D,2} \leftarrow \max(\beta_2 \gamma_{D,2}, \max_h |\hat{g}_{D,Y}[h,\cdot]|)$
        $D \leftarrow D - \frac{\kappa_D}{1-\beta_1^\tau} \cdot \frac{\gamma_{D,1}}{\gamma_{D,2}+\varepsilon}$

**Output:** $\theta, D$

---

Typical choice: $N = 70{,}000$, $\kappa_\theta = 10^{-3}$, $\kappa_D = 10^{-4}$, $\beta_1 = 0.9$, $\beta_2 = 0.999$, $\varepsilon = 10^{-7}$.

### 3.8. Resynthesis

After training, we apply the network once again on all the time frames $Z[k,\cdot]$, $k = 1,\ldots,m$. To prevent randomness in the output, rather than sampling according to $\pi_\theta(\omega_s|Y)$, we use the mode of $\pi_\theta(\nu_j,\eta_j|Y)$ and then, with $\nu_j, \eta_j$ fixed, the modes of $\pi_\theta(b_j|Y,\nu_j,\eta_j)$ and of $\pi_\theta(u_j|Y,\nu_j,\eta_j)$.

We project the thereby obtained time-frequency coefficients $z_j^{\text{dir}}[k,l]$ back into real-valued time-domain signals for each instrument via, cf. [35,39]:

$$x_j^{\text{syn}}(t) = \sum_{k,l} z_j^{\text{dir}}[k,l]\,\tilde{w}(t-\alpha k)\,e^{i2\pi\beta l(t-\alpha k)}, \tag{30}$$

where

$$\tilde{w}(t) = \frac{\beta\,w(t)}{\sum_k |w(t-\alpha k)|^2} \tag{31}$$

is the *synthesis window*. If the support of $w$ is cut to $[-1/(2\beta), 1/(2\beta)]$, then it is also the *Gabor canonical dual window* of $w$, which is the miminum-$L_2$-norm window to invert the transformation from $X$ to $Z$ according to (5).

## 4. Experimental Results and Discussion

We compare our algorithm against two other blind source separation algorithms. We selected them for their ability to identify the sound of musical instruments at arbitrary pitch on a continuous frequency axis.

1. The algorithm from a previous publication of some of the authors [20] assumes an identical tone model, but instead of a trained neural network, it uses a hand-crafted sparse pursuit algorithm for identification, and it operates on a specially computed log-frequency spectrogram. While the data model can represent inharmonicity, it is not fully incorporated into the pursuit algorithm. Moreover, information is lost in the creation of the spectrogram. Since the algorithm operates completely in the real domain, it does not consider phase information, which can lead to problems in the presence of beats. The conceptual advantage of the method is that it only requires rather few hyperparameters and their choice is not critical.

2. The algorithm by Duan et al. [18] detects and clusters peaks in a linear-frequency STFT spectrogram via a probabilistic model. Its main advantage over other methods is that it can extract instrumental music out of a mixture with signals that cannot be represented. However, this comes at the cost of having to tune the parameters for the clustering algorithm specifically for every sample.

While hyperparameter choice is more important for our new algorithm than for the first algorithm from this list, we aim to maintain our notion of blind separation by keeping our choice constant for all the samples that we consider, while for any comparison to the second algorithm, it should be kept in mind that the hyperparameters for this method are hand-optimized with the values taken from [18,20]. When comparing to the algorithm from [20], we always consider the result with spectral masking applied.

For reconstruction, unless otherwise stated, we use the values from (9). However, in training, we aim to increase the amount of training data available to the network (*data augmentation*) by using a modified time constant $\tilde{\alpha} = \alpha/4$. While the newly added spectra are completely redundant and do not add any information to the original ones, this connection is not built into the neural network and therefore the redundant training data can help it learn details that it would not learn from a representation without this redundancy.

For all the samples with 2 instruments, we train for $100,000$ iterations (independently of the size of one epoch), but we always use the result after $70,000$ iterations (*early stopping*), since it appears that this usually gives better results. We conjecture that the partially trained network itself provides a good regularization to which separations are realistic, while simply minimizing the loss itself can lead to degenerate results. Regularization via network architecture in unsupervised learning has been prominently pioneered via the deep image prior approach [26].

For assessing separation quality, we use the SDR (signal-to-distortion ratio), which measures the overall similarity between the original and the resynthesized signal, the SIR (signal-to-interference ratio), which gives the interference from the other instrument tracks in the considered signal, and the SAR (signal-to-artifacts ratio), which disregards interference and compares the given signal to all the original ones [40]. These are well-established figures in blind separation from the *BSS Eval* software package [41]. We follow the definitions compatible with version 2 of BSS Eval, which, unlike version 3, does not permit shifts in the signal. For all of them, a higher number corresponds to better quality.

When training non-trivial neural networks, it is always, to varying extent, a matter of chance if the optimization process will converge to a good value. We therefore train an *ensemble* of neural networks by running the process with 6 different random seeds (affecting the network initialization, the batch partitioning, and the random sampling of the stochastic parameters) and choose the best one in terms of mean SDR over the instruments. These

seeds are distinct from those that we used for validation and hyperparameter selection during the design process of the algorithm. In a realistic blind scenario without ground truth, this figure would not be available, but we deem it acceptable to let the user choose (for instance, by listening comparison) between a small number of different output results. Due to the aforementioned regularization by architecture, the value of the loss function is not a reliable indicator of separation quality, but it could be integrated into an end-user interface.

Similarly, the results taken from [20] are always the best-case training outcomes out of 10 runs. By contrast, the algorithm from [18] does not rely on randomness but instead on the hand-optimized hyperparameters.

### 4.1. Mozart's Duo for Two Instruments

Like in [20], we use the 8th piece from the 12 Basset Horn Duos by Wolfgang A. Mozart (K. 487) in an arrangement by Alberto Gomez Gomez for two recorders (https://imslp.org/wiki/12_Horn_Duos,_K.487/496a_(Mozart,_Wolfgang_Amadeus), accessed on 6 September 2021) as an example piece. In one sample, it is played with recorder and violin, and in the second sample, it is played with clarinet and piano. All the instruments are acoustic.

In Table 1, we compare the performance of our algorithm to that of the other two on both samples. The sample with recorder and violin is comparatively "easy." While our proposed algorithm universally gives the best SIR values (indicating low interference between the instrument tracks), the algorithm from [20] outperforms it for the recorder track in terms of SDR and SAR. Possible explanations include a more effective optimization process of the respective objective function, but it could also be a result of the different data representation used in that method (namely the special log-frequency spectrogram) or a regularizing effect of the sparse pursuit algorithm. In preliminary experiments, we found that in this particular sample, a lower value for $\mu_1$ can increase separation quality (putting more emphasis on the correctness of the direct prediction rather than the dictionary-based prediction), but requiring the user to guess the difficulty beforehand violates our notion of blind separation.

**Table 1.** Comparison of the separation algorithms on the samples based on the piece by Mozart. Best numbers are highlighted.

| Method | Instrument | SDR | SIR | SAR |
|---|---|---|---|---|
| Ours | Recorder | 13.1 | 34.8 | 13.2 |
| | Violin | 13.4 | 34.2 | 13.5 |
| | Clarinet | 12.4 | 28.0 | 12.6 |
| | Piano | 8.1 | 42.2 | 8.1 |
| [20] | Recorder | 15.1 | 32.4 | 15.2 |
| | Violin | 11.9 | 23.8 | 12.2 |
| | Clarinet | 4.1 | 24.3 | 4.1 |
| | Piano | 2.1 | 9.3 | 3.5 |
| [18] | Recorder | 10.6 | 21.4 | 11.0 |
| | Violin | 5.8 | 18.4 | 6.1 |
| | Clarinet | 6.7 | 21.3 | 6.9 |
| | Piano | 5.5 | 16.4 | 5.9 |

The first few seconds of the separation result are displayed in spectrogram form in Figure 6. Overall, the direct predictions are very accurate, but like it was observed in [20], the last tone visible in Figure 6b jumps up a fifth, which it does not do in the ground truth. This is because the recorder tone is actually one octave above the violin tone, and the overlap is virtually perfect. In such cases, since the dictionary model is never fully accurate, it may happen that an incorrect solution actually yields a lower loss.

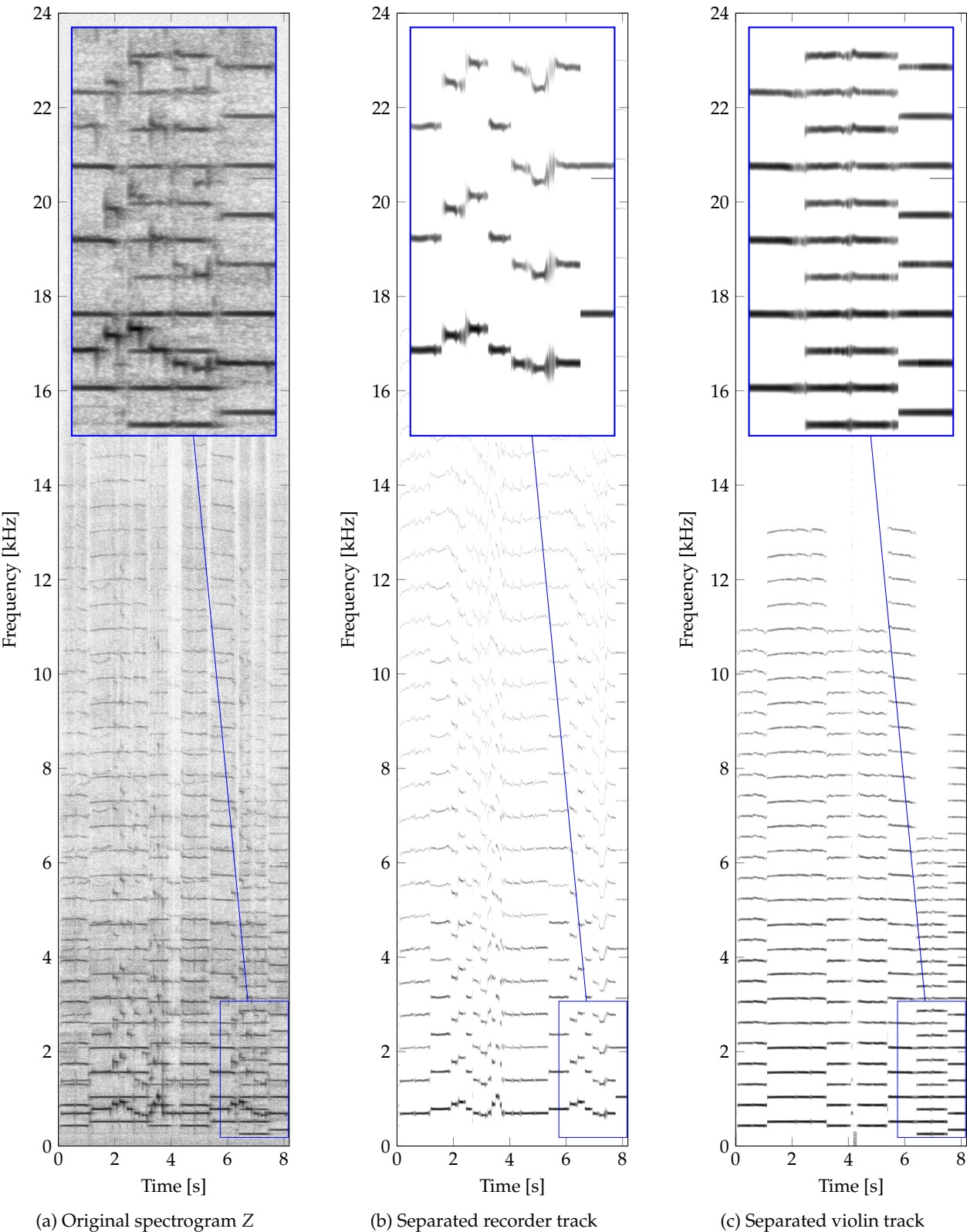

(a) Original spectrogram $Z$     (b) Separated recorder track     (c) Separated violin track

**Figure 6.** Excerpt of the separation result for the piece by Mozart, played on recorder and violin. Displayed are the original STFT magnitude spectrogram as well as the direct predictions for each instrument. In the highlighted section, the last tone is supposed to be a constant octave interval between the violin and the recorder, but the prediction for the recorder contains an erroneous jump. The color axes of the plots are normalized individually to a dynamic range of 100 dB.

By comparison, the sample with clarinet and piano is rather "hard," and our algorithm clearly shows superior performance, especially in the separation quality of the piano track. An interesting observation in Figure 7a is that while the separation performance first reaches a plateau around 10 dB (in the mean), it then declines to around 8 dB. However, as shown in Figure 7b, the values of both $d_{2,\delta}^{q,\mathrm{abs}}(Y, y^{\mathrm{spr}})$ and $d_{2,\delta}^{q,\mathrm{rad}}(Y, y^{\mathrm{dir}})$ decrease, indicating this is not an optimization failure. At the same time, the value of the regularization loss (which, with $\mu_3 = 1$, has a much lower weight than the others) increases. Therefore, this particular sample could potentially benefit from more regularization.

In Figure 8, the learning curves for both samples over all the respective runs are displayed. It is obvious that the results for the sample with recorder and violin (Figure 8a) are more consistent than those for the sample with clarinet and piano (Figure 8b). Moreover, we can see that in the former sample, separation quality generally deteriorates after too many iterations while in the latter, some runs only achieve peak performance near the end of the training. Thus, different samples can benefit from taking the results at different points in the training process. Some curves in Figure 8b terminate early, which was due to numerical failures in the training process.

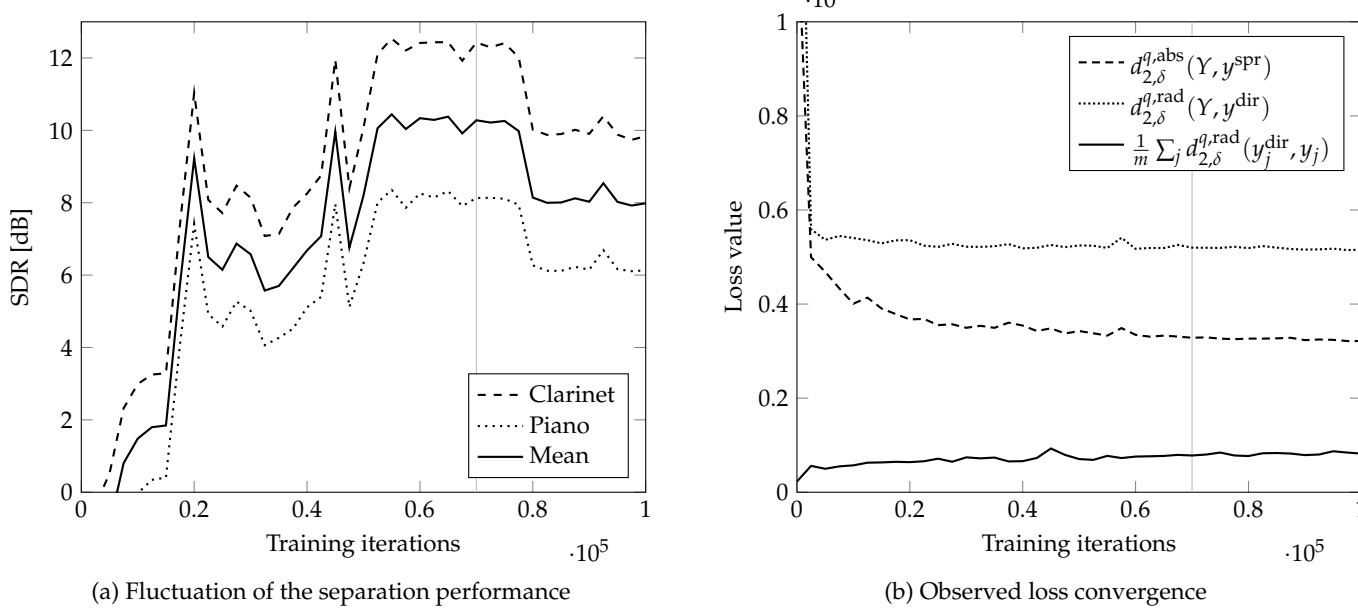

(a) Fluctuation of the separation performance

(b) Observed loss convergence

**Figure 7.** Separation performance and loss values while training on the sample with clarinet and piano in the best-case run. The vertical gray lines indicate the point at which the result was taken (70,000 iterations).

One difficulty with the piano as an instrument is that it exhibits significant inharmonicity, cf. [34]. While the algorithm from [20] has been shown to correctly identify the inharmonicity parameter on the isolated piano track, it relies on cross-correlation *without* inharmonicity for tone detection. By contrast, the algorithm presented here is based on a neural network, so correctly dealing with inharmonicity at the input stage is merely a matter of training. Since inharmonicity mostly affects higher harmonics which have low volume, the ability to represent it in the output stage does not show up as much in the $\ell_2$-based SDR/SAR/SIR figures. However, due to the lifting property of $d_{2,\delta}^{q,\mathrm{rad}}$ and $d_{2,\delta}^{q,\mathrm{abs}}$ (with $q = 0.5 < 1$), it does influence the losses.

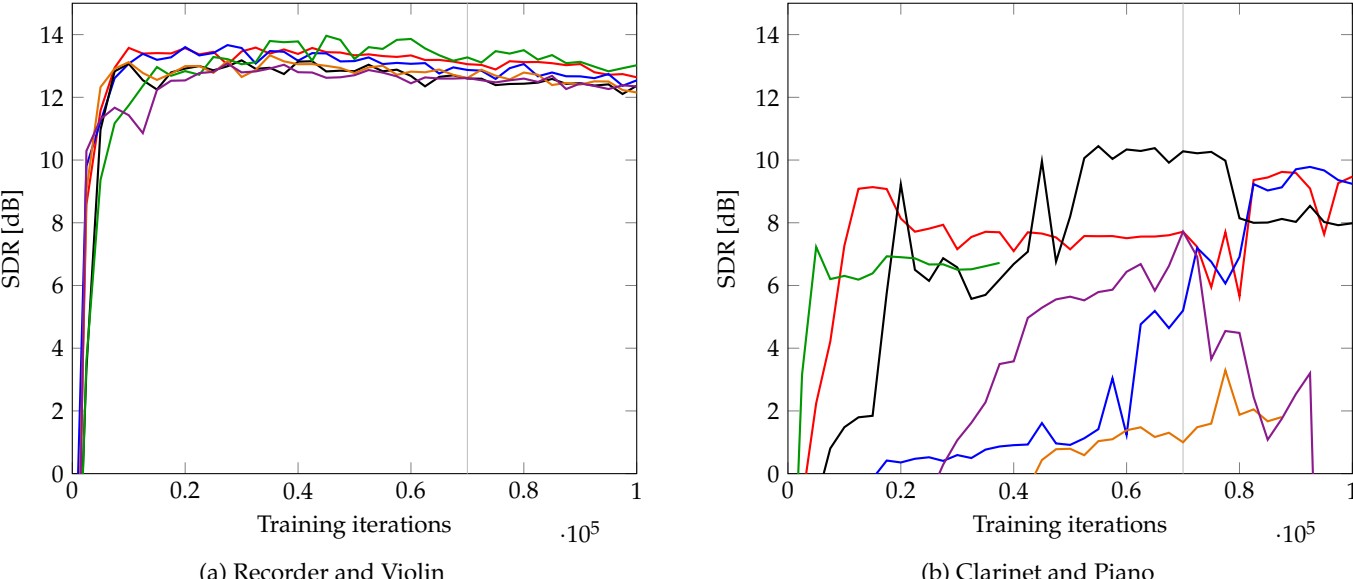

(a) Recorder and Violin

(b) Clarinet and Piano

**Figure 8.** Mean separation performance over the instruments in the samples based on the piece by Mozart. Each line represents a different run with specific random seeds. The vertical gray lines indicate the point at which the result was taken (70,000 iterations).

To illustrate the effect, we also ran our algorithm on the isolated piano track, once with and once without the inharmonicity parameter in the model, with 4 distinct random seeds each. The results for the random seeds of 11 are displayed in Figure 9. While the difference appears small on a global scale, it is consistent.

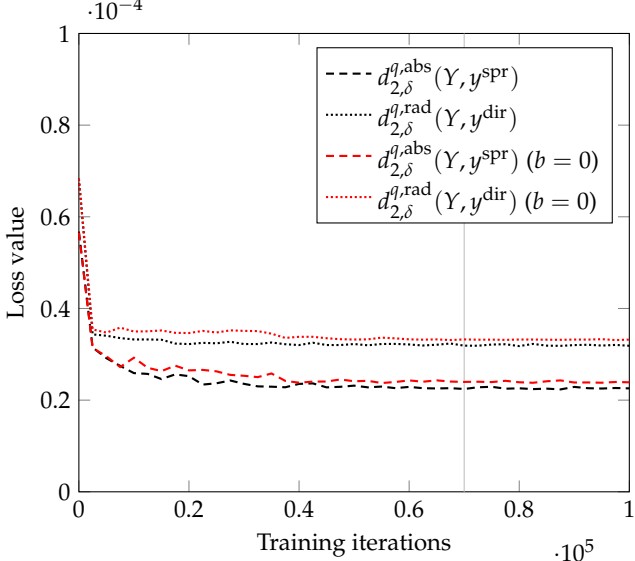

**Figure 9.** Observed loss convergence on the isolated piano track.

*4.2. URMP*

The URMP dataset [42] consists of samples with two or more acoustic instruments. It was not created for blind separation, so the samples are generally too "hard" to be used in that context. Nevertheless, in [20], a subset of potentially appropriate samples was determined, and we compare the performance of our new algorithm on these samples.

As can be seen in Table 2, the results vary widely. For the first and the fourth sample, we have to declare a failure compared to the other two algorithms. On the second and the third sample, however, our algorithm is universally dominant. The good performance on the sample with trumpet and saxophone is especially surprising, since we deemed this a very challenging sample due to the similarity of the sounds of the instruments.

**Table 2.** Comparison of the separation algorithms on a selection of samples from the URMP [42] dataset. Best numbers are highlighted.

| Method | Instrument | SDR | SIR | SAR |
|:---:|:---:|:---:|:---:|:---:|
| Ours | Flute | −4.7 | 17.5 | −4.6 |
| | Clarinet | 5.0 | 10.1 | 7.0 |
| | Trumpet | 7.7 | 19.9 | 8.0 |
| | Violin | 9.7 | 30.7 | 9.7 |
| | Trumpet | 8.4 | 30.3 | 8.4 |
| | Saxophone | 13.0 | 24.9 | 13.3 |
| | Oboe | 2.9 | 6.9 | 5.9 |
| | Cello | −0.6 | 19.2 | −0.5 |
| [20] | Flute | 2.4 | 9.5 | 3.9 |
| | Clarinet | 6.2 | 25.3 | 6.3 |
| | Trumpet | 5.3 | 16.6 | 5.7 |
| | Violin | 7.7 | 25.1 | 7.8 |
| | Trumpet | −2.4 | 1.1 | 2.7 |
| | Saxophone | 0.1 | 22.5 | 0.2 |
| | Oboe | 6.3 | 17.0 | 6.8 |
| | Cello | 4.2 | 17.1 | 4.5 |
| [18] | Flute | 3.4 | 19.6 | 3.6 |
| | Clarinet | 2.1 | 5.9 | 5.4 |
| | Trumpet | — | — | — |
| | Violin | — | — | — |
| | Trumpet | 1.2 | 9.4 | 2.3 |
| | Saxophone | 6.9 | 17.2 | 7.4 |
| | Oboe | −0.8 | 13.1 | −0.4 |
| | Cello | 3.4 | 6.4 | 7.3 |

Oracle Dictionary

In order to investigate the failure of the separation method on the first and the fourth sample, we first train *oracle* dictionaries by providing the algorithm with the ground-truth individual tracks for the respective instruments. We then supply the separation procedures with these dictionaries as initial values. In one instance, we keep the dictionaries fixed throughout the training, and in another one, we train then at the normal rate, starting from oracle dictionaries. For each separation, we use 4 different random seeds. The results are displayed in Table 3. While we usually use the direct prediction $z^{\text{dir}}[k, l]$ for resynthesis, we here also include the resynthesis based on the dictionary prediction $z[k, l]$ for analysis.

With the fixed oracle dictionary, the results are generally much better than with the normal training in Table 2. However, when allowing training from the oracle dictionary, the separated flute and the cello tracks become unacceptably bad again. When using the dictionary prediction for resynthesis, all the results are of very poor quality, indicating that the dictionary model (10) is not an accurate representation of the spectral characteristics of the tones.

**Table 3.** Separation with an oracle dictionary on a selection of samples from the URMP [42] dataset. The "Fix" column indicates whether the dictionary is kept constant during the separation, and the "Pred." column specifies whether the direct or the dictionary prediction is used. Best numbers are highlighted when they also exceed the performance from Table 2.

| Fix | Pred. | Instrument | SDR | SIR | SAR |
|-----|-------|-----------|-----|-----|-----|
| Yes | Dir. | Flute | 1.2 | 9.4 | 2.4 |
| | | Clarinet | 5.7 | 25.7 | 5.8 |
| | | Oboe | 5.3 | 11.2 | 6.8 |
| | | Cello | 3.0 | 30.3 | 3.0 |
| | Dict. | Flute | −0.5 | 1.0 | 0.3 |
| | | Clarinet | 1.8 | 30.2 | 1.8 |
| | | Oboe | 0.5 | 9.6 | 1.6 |
| | | Cello | −1.4 | 25.4 | −1.4 |
| No | Dir. | Flute | −0.4 | 21.6 | −0.3 |
| | | Clarinet | 7.0 | 13.2 | 8.4 |
| | | Oboe | 3.7 | 8.0 | 6.3 |
| | | Cello | 0.4 | 26.1 | 0.5 |
| | Dict. | Flute | −5.1 | 24.6 | −5.1 |
| | | Clarinet | 2.2 | 17.3 | 2.4 |
| | | Oboe | −1.8 | 4.6 | 0.6 |
| | | Cello | −2.8 | 23.7 | −2.8 |

Upon manual inspection of the flute track, we noticed that it contains a number of tones which are "half-overblown," such that the spectra of both the higher octave and of the lower octave are present. This does not represent the normal spectral characterics of the flute sound, so the oracle dictionary contains a "compromise," while the trained dictionary fails to represent these half-overblown tones. In the cello track, there is no obvious technical peculiarity, but the tones are simply very diverse, involving different open strings and also different articulation between the tones, so even the training of the oracle dictionary is problematic.

Generally, for instruments with inconsistent spectral characterics, the method from [20] may be at an advantage since it prunes and randomly reinitializes parts of the dictionary in regular intervals; so, given enough tries, it can reach an appropriate dictionary by chance, even if it is potentially suboptimal with respect to the loss function.

*4.3. Duan et al.*

In [18], a number of original samples are used. As we mentioned, their algorithm has the unique ability of separating a representable instrument track out of a mixture with a non-representable residual, which can, for instance, be a singing voice. Since our algorithm is not designed for such signals, we selected the samples for which all the instruments can be represented. In total, these are one sample with acoustic oboe and euphonium, one sample with *synthetic* piccolo and organ, and a third sample with a synthetic oboe track added to the previous sample.

The problems with these samples are that they have a different sampling frequency ($f_s = 22.05$ kHz) and they are also very short. Whereas in [20], the signals were converted to a different sampling frequency as a preprocessing step in order to reduce the loss of resolution due to smoothing, we do not have this problem here. While we keep the value of $\zeta$ from (9) constant in terms of absolute units, the relation to the sampling frequency consequently changes to $\zeta f_s = 470.4$. The frequency constant changes proportionately with the sampling frequency to $\beta = 1.794\,433\,593\,75$ Hz. Since the samples are short, we choose an even smaller time constant compared to (9) by setting $\alpha = 16/f_s \approx 0.73$ ms in the acoustic sample and $\alpha = 128/f_s \approx 5.80$ ms in the synthetic ones, each with $\tilde{\alpha} = \alpha/4$ for training. The results are displayed in Table 4.

**Table 4.** Comparison of the separation algorithms on the data by [18]. Instruments labeled as "s." are synthetic, those labeled as "a." are acoustic. Best numbers are highlighted.

| Method | Instrument | SDR | SIR | SAR |
|---|---|---|---|---|
| Ours | Oboe (a.) | 9.6 | 47.2 | 9.6 |
| | Euphonium (a.) | 8.7 | 33.7 | 8.7 |
| | Piccolo (s.) | 17.2 | 36.5 | 17.2 |
| | Organ (s.) | 14.3 | 50.3 | 14.3 |
| | Piccolo (s.) | 6.8 | 22.1 | 6.9 |
| | Organ (s.) | 7.3 | 19.2 | 7.7 |
| | Oboe (s.) | 8.3 | 46.3 | 8.3 |
| [20] | Oboe (a.) | 18.6 | 33.6 | 18.8 |
| | Euphonium (a.) | 14.7 | 31.5 | 14.7 |
| | Piccolo (s.) | 11.2 | 25.9 | 11.3 |
| | Organ (s.) | 10.1 | 20.7 | 10.5 |
| | Piccolo (s.) | 4.2 | 24.8 | 4.3 |
| | Organ (s.) | 6.0 | 20.0 | 6.3 |
| | Oboe (s.) | 5.3 | 12.4 | 6.4 |
| [18] | Oboe (a.) | 8.7 | 25.8 | 8.8 |
| | Euphonium (a.) | 4.6 | 14.5 | 5.3 |
| | Piccolo (s.) | 14.2 | 27.9 | 14.4 |
| | Organ (s.) | 11.8 | 25.1 | 12.1 |
| | Piccolo (s.) | 6.5 | 20.0 | 6.7 |
| | Organ (s.) | 6.6 | 17.3 | 7.1 |
| | Oboe (s.) | 9.0 | 21.9 | 9.2 |

While the results for the acoustic sample are better than in the original publication, they are still not nearly as good as those in [20]. Our explanation is that while the time resolution of the spectrogram is almost as high as that of the time-domain signal ($\tilde{\alpha} = 4/f_s$), there is still just not enough data in the sample to train the neural network, and thus hand-crafted methods are at an advantage.

By contrast, our method delivers very good results with synthetic instruments, clearly and universally outperforming the other methods on the sample with two instruments and providing the best average performance on the sample with three instruments.

## 5. Conclusions

We have developed a blind source separation method that unmixes the contributions of different instruments in a polyphonic music recording via a parametric model and a dictionary. The model parameters are predicted by a deep convolutional neural network, and with respect to those that do not possess a useful backpropagation gradient, we use the policy gradient instead.

Unlike other algorithms, ours operates directly on the complex output of the STFT, which is linear and preserves the phase. Rather than using spectral masking, we let the network give a direct prediction for the complex amplitudes of the harmonics.

In general, the algorithm exhibits very good performance on a variety of samples. It is especially dominant in terms of SIR, which is relevant since eliminating cross-talk is the main objective in separation. We attribute this to the use of a complex-valued direct prediction for the individual instruments that can properly handle interference between the instrument tones in the spectral domain. Such interference is particularly prevalent in synthetic samples, on which the performance of our algorithm surpasses that of the competing methods. It also clearly outperforms the other methods on the sample with the acoustic piano, which poses the challenge of detecting tones with inharmonicity.

As is usual with blind separation, however, problems arise when the structural assumptions are not satisfied. In two samples from the URMP database, there was one instrument each whose sound in the recording varied too much to be accurately represented by the dictionary. While we found that using oracle dictionaries, satisfactory

separation can be achieved, these dictionaries are not attained as the result of training, even when they are supplied as the initial value.

Due to the use of neural networks, our approach is very flexible with respect to the choice of the loss function. In the spirit of blind separation, we chose the weights for the respective distances such that they constitute a reasonable comprise for all the samples on which we tested them, but this choice is not necessarily optimal for the individual samples. While a linear combination is the most straight-forward way to account for all the considered distances, a non-linear mapping could potentially be better. Moreover, even though we have not found any distance functions yielding better performance than the ones we use, more experiments could be conducted.

With our approach, we hope to provide a blueprint for the combined use of backpropagation gradients and policy gradients in the application of neural networks on non-convex parameter identification problems.

**Author Contributions:** S.S. devised, implemented, and tested the algorithm. S.S. and J.L. wrote the manuscript and conceived additional experiments. E.J.K. supervised the research and revised the manuscript. All authors have read and agreed to the published version of the manuscript

**Funding:** S.S. and J.L. acknowledge funding by the Deutsche Forschungsgemeinschaft (DFG, German Research Foundation)–project number 281474342/GRK2224/1.

**Institutional Review Board Statement:** Not applicable.

**Informed Consent Statement:** Not applicable.

**Data Availability Statement:** The functioning source code for the algorithm is available on GitHub (https://github.com/rgcda/Musisep, accessed on 6 September 2021) under the GNU General Public License (version 3). The input data from [18] can be downloaded from the respective website (https://sites.google.com/site/mperesult/musicseparationresults, accessed on 6 September 2021). For all the other samples, we provide the input data along with the best-case separation results on the institute website (https://www.math.colostate.edu/~king/software.html#Musisep, accessed on 6 September 2021). Due to the stochastic nature of parallel computing, training results are not exactly reproducible.

**Acknowledgments:** The authors would like to thank Sören Dittmer and Louisa Kinzel for sharing their expertise and ideas on the learning approach. Further, we thank the anonymous reviewers, whose valuable and constructive comments helped us improve the manuscript.

**Conflicts of Interest:** The authors declare no conflict of interest.

## Abbreviations

The following abbreviations are used in this manuscript:

| | |
|---|---|
| DIP | Deep image prior |
| GAN | Generative adversarial network |
| MCTS | Monte Carlo tree search |
| NMF | Non-negative matrix factorization |
| PLCA | Probabilistic latent component analysis |
| REINFORCE | Reward increment = Nonnegative factor × Offset reinforcement |
| | × Characteristic eligibility |
| SAR | Signal-to-artifacts ratio |
| SDR | Signal-to-distortion ratio |
| SIR | Signal-to-interference ratio |
| STFT | Short-time Fourier transform |
| URMP | University of Rochester Multi-Modal Music Performance Dataset |

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
