# Peer review of "Blind Source Separation in Polyphonic Music Recordings Using Deep Neural Networks Trained via Policy Gradients"

_signals_

Round 1
Reviewer 1 Report
This paper presents a new method for the blind source separation of musical instruments in audio signals of musical pieces. Source separation is achieved via reconstructing a time-frequency representation of the signal using a parametric instrument model. The method is hybrid in the sense that parameters are predicted using a neural network. Overall, the approach appears plausible. It is commendable that the authors provided the source code of the method. Clearly, the method works very well, as demonstrated by the online demo and API. Lines 171 and 172 read, (the loss is non-convex) "Therefore, conventional optimization methods based on gradient descent are not a good choice for minimizing the ". My main technical concern is related to this statement. My intuition is that RL adds significant complexity to training; furthermore, most modern deep learning optimization algorithms are well suited for optimizing non-convex loss functions, particularly when approaching the mean-field limit. Here, the hybrid approach complicates this, but even so, have the authors tried end-to-end training using backpropagation only? Did it prove to be more inefficient than RL? Can it be proven empirically that it is worth removing these parameters from backprop and practically introducing and outer loop of RL? If so, is RL superior to e.g., a genetic algorithm? If it is the hybrid reconstruction loss that complicates things, can a better loss function be found to avoid RL? I suggest adding more discussion and empirical evidence to clarify this. I recommend adding illustrations to better explain the method, i.e., to visually illustrate the signals of the tone model, adding a figure of a representative STFT, etc. In my opinion, the title places too much emphasis on *training* the neural network, whereas the manuscript really describes a novel method, not only its training. Therefore, the authors might consider changing the title. I recommend a minor, stylistic revision of the language, preferably by a native speaker. Other than these, the paper is technically excellent, well-written and presents a relevant and novel solution; therefore, I recommend publication.Author Response
Please see the attachment.

Reviewer 2 Report
This paper proposes a method for blind separation of sounds of musical instruments in audio signals. The process involves examining individual tones via a parametric model, training a dictionary to capture the relative amplitudes of the harmonics. The work uses a U-Net deep neural network trained without ground truth information. The algorithm yields good separation with low interference on several audio samples, if given enough data for the training.
The paper is easy to read, in a good writing style.
The work is somewhat distinct from my expertise (speech processing). I reviewed it on a general basis, and cannot vouch for accuracy in the math.
Specific points:
.. individual STFT time frames… - STFT is not defined, but assumed; worse, the spelled-out version occurs in the 9th line of the text without noting “STFT”
..plays a single tone at a time ..- technical papers in this field should be precise when using terms such as “tone”; often “tone” means a sinusoid, which is not what an instrument usually yields; I assume here that tone means a periodic sound with a single fundamental; if so, say that
..Gabor frame ..- while this term is found in some practical mathematical work, it is surely worth defining it here, as most audio papers do not use it
The text seems haphazard in use of “cf.”, i.e., inside brackets for ref. 2, outside for refs. 3-5.
..A large class of algorithms for this problem are ..
..Many algorithms for this problem are ..
..In the simplest form, each tone of an instrument at a particular pitch has its own representation [6,7]. - this is not informative; explain the representation; do not force the reader to read the references
..To make the representation of the sound of a particular instrument valid .. - what does this mean? i.e., what makes it valid?
..one often employs tensor factorization .. - and what is this?
..Decreasing the variability in a model .. - does this mean use fewer parameters?
..bottom-up physical model for the tones of the instruments .. - what is this? So far, this paper uses lots of technical terms and jargon, with little explanation of these
..be independent of the discretization of the frequency axis such that tones outside of strict equal-temperament tuning .. - this is another example of lots of jargon: 1) how does discretization of the frequency axis affect anything relevant here? 2) what effect does independence have? 3) what is equal-temperament tuning? 4) what effect do tones outside have?
..log-frequency spectrogram can still be helpful to ease the detection of tones .. - in what way does the spectrogram affect this process? What does “ease the detection” mean? Indeed, what is the process of this detection?
..For purely sinusoidal signals, the spectrum is modeled precisely by (8), ..- be more precise; a truly “purely sinusoidal signal” is a simple sinusoid, for which (8) is not needed; and if one means “a signal consisting only of sinusoids,” then one needs infinitely long signals
..network for the follwing steps. ->
..network for the following steps.
- .. wind and string instruments
.. – why not others? E.g.. brass - .. over longer periods of time;
.. – longer than what? Be specific
- .. it as a stochatic parameter
.. -> - .. it as a stochastic parameter
..At infinity, it always decays exponentially,
.. – one never gets “at infinity”
.. for each tone are are .. ->
.. for each tone are ..
..Finally, we add two more convolutional layers with 80 filters of sizes .. - are there any specific motivations for all this parametric choices?
..better than in original publication, .. ->
..better than in the original publication, ..
..functions yielding to better performance ..
..functions yielding better performance ..
Several of the references have repetitive information, e.g., dates in .. Interspeech 2020, Shanghai, China, 25–29 October 2020, 2020
Reviewer 3 Report
This paper makes a solid contribution in blind source separation applied to musical instruments. The paper extends the research reported in [18]. Comparisons are made against the methods in [17] and [18] using publicly available benchmark datasets. Another strong point is the release of code.
The major weakness of the paper is in the writing style which ignores the reader, because there is not smooth transition from one step to the other in the analysis. I will give some examples:
First of all, there is no need to refer to wave propagation. One may start from (2) which is simply the synthesis equation of a continuous-time Fourier series. Consider if $x(t)$ should be replaced by $X(t)$ as in (4). Justify (5). Check if some constants were omitted in (7). Explain why $-i 2\pi \beta l$ is omitted in (8). Explain (9). Explain Y[l] and y[l] in (12).
An example of good derivation is (15).
Explain how $\varpi_{\theta}^{R_i}$ are obtained in (28).
SDR, SIR, SAR could be formally defined.
One can hardly identify any trend of convergence in Figure 6b.
I would also suggest to remove minor details and technicalities in order to gain space to demonstrate the major research outcomes.
Round 2
Reviewer 3 Report
The authors have addressed this Reviewer comments adequately. The manuscript quality and readability has improved.